# Morin Sensitizes MDA-MB-231 Triple-Negative Breast Cancer Cells to Doxorubicin Cytotoxicity by Suppressing FOXM1 and Attenuating EGFR/STAT3 Signaling Pathways

**DOI:** 10.3390/ph16050672

**Published:** 2023-04-29

**Authors:** Sushma Maharjan, Min-Gu Lee, So-Young Kim, Kyu-Shik Lee, Kyung-Soo Nam

**Affiliations:** Department of Pharmacology, College of Medicine and Intractable Disease Research Center, Dongguk University, Gyeongju 38066, Republic of Korea; maharjan.sushma@gmail.com (S.M.); mklee@dongguk.ac.kr (M.-G.L.); soyoungkim@dongguk.ac.kr (S.-Y.K.)

**Keywords:** morin, MDA-MB-231 triple-negative breast cancer (TNBC), doxorubicin, synergistic effect, FOXM1, EGFR/STAT3

## Abstract

Considerable emphasis is being placed on combinatorial chemotherapeutic/natural treatments for breast cancer. This study reveals the synergistic anti-tumor activity of morin and Doxorubicin (Dox) co-treatment on MDA-MB-231 triple-negative breast cancer (TNBC) cell proliferation. Morin/Dox treatment promoted Dox uptake and induced DNA damage and formation of nuclear foci of p-H2A.X. Furthermore, DNA repair proteins, RAD51 and survivin, and cell cycle proteins, cyclin B1 and forkhead Box M1 (FOXM1), were induced by Dox alone but attenuated by morin/Dox co-treatment. In addition, Annexin V/7-AAD analysis revealed that necrotic cell death after co-treatment and apoptotic cell death by Dox alone were associated with the induction of cleaved PARP and caspase-7 without Bcl-2 family involvement. FOXM1 inhibition by thiostrepton showed that co-treatment caused FOXM1-mediated cell death. Furthermore, co-treatment downregulated the phosphorylation of EGFR and STAT3. Flow cytometry showed that the accumulation of cells in the G2/M and S phases might be linked to cellular Dox uptake, p21 upregulation, and cyclin D1 downregulation. Taken together, our study shows that the anti-tumor effect of morin/Dox co-treatment is due to the suppression of FOXM1 and attenuation of EGFR/STAT3 signaling pathways in MDA-MB-231 TNBC cells, which suggests that morin offers a means of improving therapeutic efficacy in TNBC patients.

## 1. Introduction

Combinatorial cancer regimens have been investigated extensively for more than four decades to overcome the high failure rate of single chemotherapy and improve treatment efficacy and quality of life [1]. Adjuvant breast cancer treatments employ different types of anti-cancer drugs, such as Doxorubicin, paclitaxel, and cyclophosphamide [2]. Doxorubicin (Dox) is a potent anthracycline used to treat breast cancer and a first-line therapy for triple-negative breast cancer (TNBC) [3]. Despite the remarkable anti-tumor activity, clinical applications of Dox are limited by its acute and chronic toxicities, which include myelosuppression, immunosuppression, and cumulative cardiotoxicity [4]. For example, cardiomyopathy develops in 50–60% of patients receiving high doses of Dox, which can limit treatment [5]. Thus, combinatorial chemotherapy offers a potential means of circumventing the dose-dependent adverse events of chemotherapeutics [6].

TNBC is a heterogeneous group of aggressive tumors that constitute ~15% of all breast cancers [7]. Treatment of TNBC is more difficult than other subtypes of breast cancer because it lacks the ER, PR, and HER2 receptors [8]. For this reason, TNBC patients are typically treated with surgery, conventional chemotherapy, and radiotherapy [9]. Doxorubicin-based therapies have been used to treat TNBC and other cancers for several decades, but dose-dependent side effects limit their use [10]. Several approaches have been used to overcome this limitation, such as dosage optimization, the use of analogs, and combined therapy [11]. Recently, there has been interest in combination therapies that use relatively non-toxic phytochemicals and chemotherapeutics, as several studies have shown that many cytotoxic plant extracts and phytochemicals work in synergy with Doxorubicin [12,13].

Morin is a widely available phytochemical with anti-cancer, anti-inflammatory, and cardiovascular protective activities in various cancers [14,15,16]. Furthermore, morin exhibits anti-cancer activity in different cancer cells by controlling genes related to cell growth and survival, inducing apoptosis, and causing chemosensitization [17,18]. Morin can also enhance the cytotoxicity of chemotherapeutics in cancer cells [19,20]. However, the effect of co-treatment of morin and Dox in MDA-MB-231 cells has not been investigated. Therefore, we assessed the effect of morin/Dox co-treatment in MDA-MB-231 cells to determine whether morin could enhance the anti-tumor activity of Dox and minimize Dox-related toxicities.

## 2. Results

### 2.1. Morin/Dox Co-Treatment Increased the Cytotoxic Effect on MDA-MB-231 TNBC Cells

SRB assay showed a significant decrease in MDA-MB-231 cell viability after Dox treatment, whereas morin caused less reduction in cell viability (Figure 1A,B). Interestingly, co-treatment of Dox (0.3 μM) and morin (0–100 µM) significantly decreased cell viabilities (Figure 1C). In addition, the IC50 values of morin/Dox and Dox were 0.25 and 0.58, respectively. CompuSyn software version 1.0 (Ting Chao Chou and Nick Martin, Paramus, NJ, USA) was used to demonstrate the drug synergism, whereas CIs < 1 (Figure 1D), indicating morin/Dox co-treatment synergistically reduced cell viabilities. Furthermore, cellular Dox uptake by MDA-MB-231 cells was enhanced by morin/Dox co-treatment (Figure 1E). These observations indicate that morin/Dox co-treatment markedly decreases cell viability by increasing the cellular uptake of Dox.

### 2.2. Morin/Dox Co-Treatment Increased DNA Damage and p38 Activation

Dox is a potent DNA-damaging agent, causes phosphorylation of H2A.X and the formation of discrete nuclear foci [21]. This damage triggers the activation of the p38 MAPK signaling pathway, which ultimately results in cell death [22]. We observed morin/Dox co-treatment phosphorylation of H2A.X and formation of nuclear foci (Figure 2A,B), activated p38 MAPK (Figure 2A), and slightly activated ERK but had no effect on JNK (Figure 2C). These findings suggest that the p38 MAPK pathway activation may involve in morin/Dox-induced DNA damage.

### 2.3. Morin/Dox Co-Treatment Promoted Necrotic Cell Death

Dox induces DNA damage and apoptotic cell death by increasing the activity of PARP and caspase executioners such as caspases-3 and -7 [23]. The treatment with Dox alone induced cleavages of PARP and caspase-7, but morin/Dox co-treatment did not enhance the levels of cleaved PARP and cleaved caspase-7 (Figure 3A). Additionally, flow cytometry analysis showed that while Dox induced apoptosis, the inclusion of morin promoted necrosis (Figure 3B). Interestingly, co-treatment did not affect the protein levels of Bcl-xL, Bcl-2, and BAX (Figure 4A), but the protein levels of LC-3-II (an autophagy marker) and RIPK3 (a necroptosis protein marker) were reduced by co-treatment (Figure 4B). These results suggest that morin/Dox co-treatment induces cell death predominantly through necrosis.

### 2.4. Morin/Dox Co-Treatment Induced Cell Death by Suppressing the FOXM1 Pathway

FOXM1 is a crucial transcription factor involved in various biological processes, including DNA damage repair, cell growth and, progression through cell cycle [24]. In addition, FOXM1 regulates downstream proteins, including survivin, RAD51, which contribute to homologous recombination DNA repair in breast cancer cells and, cyclin B1, a cell cycle regulatory protein [25,26]. Interestingly, treatment with Dox alone induced FOXM1, RAD51, survivin, and cyclin B1, whereas combination with morin attenuated levels of these proteins (Figure 5). Therefore, we speculated that FOXM1 might regulate these RAD51, survivin, and cyclin B1, and thus, inhibit cell proliferation, disrupt the DNA repair mechanism, and induce cell cycle arrest.

To determine whether cell death by a FOXM1 signaling pathway in MDA-MB-231 cells after morin/Dox co-treatment, thiostrepton (a specific FOXM1 inhibitor) was used. The results showed the decrease in cell viability in a concentration-dependent manner (Figure 6A). Furthermore, there was a decrease in the protein levels of FOXM1, RAD51, survivin, and cyclin B1 with thiostrepton treatment (Figure 6B). Co-treatment with Dox and thiostrepton also suppressed levels of these proteins (Figure 6C), which suggests that inhibition of the FOXM1 pathway may disrupt the homologous recombination DNA repair system, impede cell cycle progression and result in cell death. Taken together, these results support our hypothesis that FOXM1 pathway inhibition may play a critical role in morin/Dox-induced cell death.

### 2.5. Morin/Dox Co-Treatment Induced Cell Death by Inhibiting the EGFR/STAT3 Pathway

EGFR is frequently overexpressed in TNBC, and phosphorylated EGFR is among the most important downstream effectors of cell survival due to the nuclear translocation of STAT3, downstream target of EGFR [27]. When we evaluated the effects of morin and morin/Dox co-treatment on EGFR and STAT3, we found morin decreased phospho-EGFR and phospho-STAT3 without affecting total EFGR and STAT3 (Appendix A), while co-treatment concentration dependently attenuated levels of phospho-EGFR, EGFR, phospho-STAT3, and STAT3 in MDA-MB-231 cells (Figure 7), which indicates morin/Dox co-treatment inhibited cell growth and subsequent cell death might be associated with another signaling pathway via suppressing the EGFR/STAT3 signaling pathway.

### 2.6. Morin/Dox Co-Treatment Disrupted Cell Cycle Progression

When mammalian cells are exposed to DNA-damaging agents, they activate cell cycle checkpoints. This study investigated the effect of morin/Dox co-treatment on cell cycle arrest using flow cytometry. Treatment with Dox alone 0.3 µM for 24, 48, and 72 h resulted in accumulation of cells in the G2/M phase and a reduction in the proportion of cells in the S and G0/G1 phases (Figure 8A). The co-treatment of morin and Dox at 50 µM/0.3 µM enhanced the G2/M arrest and slightly attenuated the S phase population. Furthermore, at a higher concentration of morin (100 µM), the co-treatment increased the percentage of cells at G2/M phase whereas, reduced in the G0/G1 phase (Figure 8A). Overall, the results suggest that morin/Dox co-treatment enhances G2/M arrest and has a minimal effect on S arrest.

### 2.7. Morin/Dox Co-Treatment Affected the Expressions of Cell Cycle Regulators

P21 (a cyclin-dependent kinase (CDK) inhibitor) is a key cell cycle regulator and also regulates cell death [28]. Morin/Dox co-treatment increased p21 expression (Figure 8B) and significantly decreased cyclin D1 levels in MDA-MB-231 cells (Figure 8B). Cyclin D1 with CDKs promotes cell cycle progression, and its overexpression is linked to the development of cancer [29]. Consequently, these data depict the impediment of cell cycle progression related to the increase in p21 and decrease in cyclin D1 protein levels after morin/Dox co-treatment.

## 3. Discussion

Despite the potential of new biological agents for targeting triple-negative breast cancer (TNBC), chemotherapy, specifically combination therapies that include Dox, remains the primary treatment option. However, the effectiveness of Dox can be hindered by drug resistance and dose-related toxicities [8,30]. Recent studies have suggested that combining natural substances with chemotherapeutic drugs can enhance their anti-cancer effects while reducing chemoresistance and systemic toxicity [31]. Morin, a flavonoid with reported anti-cancer activity, was tested in combination with Dox on MDA-MB-231 cells to determine if it could enhance the cytotoxic effects of Dox and mitigate its adverse effects [16]. The results showed that co-treatment with morin and Dox increased the cytotoxicity of Dox, enhanced its uptake, and synergistically augmented its cytotoxicity (Figure 1). These findings suggest that combining morin with Dox could be a promising therapeutic approach for TNBC treatment.

Morin/Dox co-treatment induced the phosphorylation of the histone variant H2A.X and p38 MAPK in MDA-MB-231 cells (Figure 2A), and the phosphorylation of H2A.X is considered a marker of DNA damage [32]. In mammalian cells, p38 MAPK activation after DNA damage leads to cell death [22]. Taken together, these results suggested morin/Dox co-treatment enhanced DNA damage and led to p38 MAPK activation-induced cell death.

Dox causes DNA double-strand breaks and promotes apoptosis by reducing mitochondrial membrane potential, which activates caspases [33]. Aberrant Bcl-2 family protein levels on mitochondrial membranes serve as a marker of apoptosis [34]. Dox induced cleavage of PARP and caspase-7 while combining Dox with morin did not augment the levels of these cleaved proteins (Figure 3A). Flow cytometry analysis showed that single Dox treatment induced apoptosis and the addition of morin promoted necrosis (Figure 3B). Interestingly, combining morin with Dox did not affect the protein levels of Bcl-2 family (Figure 4A) whereas, LC3-II and RIPK3 protein levels were moderately reduced (Figure 4B). RIPK3 is involved in necroptosis, an inflammatory form of cell death [35]. Despite the decrease in RIPK3 expression, the addition of morin with Dox still increased necrotic cell death in the breast cancer cells. However, necrosis unprogrammed cell death that occurs in response to chemical or physical insult [36] and a study revealed that morin induced DNA degradation in murine hepatoma cells through necrosis or apoptosis [37]. Therefore, the data suggest that the combination treatment promoted double-strand DNA breakage and subsequent necrotic cell death, without enhancing Dox-induced apoptosis. Autophagy and necroptosis did not contribute to the observed cell death.

In response to DNA damage, cells activate DNA damage response pathways by using various transcription factors, with FOXM1 playing a critical role in regulating the transcription of DNA repair genes [38]. Research has shown that several downstream targets of FOXM1, including survivin, cyclin B1, S-Phase Kinase-Associated Protein 2, and CDC25B, were significantly upregulated in non-small-cell lung cancer cells treated with gefitinib [39]. Additionally, the DNA repair protein RAD51 is a potential downstream target of FOXM1 [40]. Furthermore, FOXM1 can activate the cyclin B1 promoter site and stall cell cycle progression [41]. Tan et al. showed significant inhibition of FOXM1 suppressed MDA-MB-231 cell tumorigenesis in in vivo [42], and silencing of FOXM1 enhanced sensitivity to cisplatin, Dox, and paclitaxel in several cancers and induced cell death [43,44]. Moreover, knockdown of cyclin B1 inhibited breast cancer cell proliferation by sensitizing them to chemotherapeutics [45]. Therefore, it can be speculated that FOXM1 is directly correlated with DNA repair and cell cycle progression. In this study, morin/Dox co-treatment reduced the Dox-induced expressions of FOXM1, RAD51, survivin, and cyclin B1 (Figure 5). To determine whether cell death is caused by the FOXM1 pathway, we used thiostrepton, a FOXM1 inhibitor, which decreased cell viability in concentration-dependent manner (Figure 6A). Furthermore, thiostrepton reduced the levels of FOXM1, RAD51, survivin, and cyclin B1 (Figure 6B) and suppressed the Dox-induced increase in these proteins as shown in Figure 6C. Therefore, it can be postulated that the attenuation of FOXM1 by co-treatment might be related to Dox sensitivity, leading to inhibition of the DNA repair system, halting cell cycle progression, and subsequent cell death. These results suggest that the FOXM1 pathway might be a major signaling pathway that induces cell death by inhibiting the DNA repair mechanism and cell cycle progression.

Next, EGFR is a transmembrane protein with tyrosine kinase activity and modulates the DNA repair mechanism via an association with the catalytic subunit of DNA protein kinase. EGFR overexpression promotes unregulated growth, inhibits apoptosis, and probably contributes to drug resistance [46]. Furthermore EGFR is the upstream target of STAT3 and is often overexpressed in TNBC [47]. However, Zhang et al. revealed that the anti-tumor activity of afatinib on intrahepatic cholangiocarcinoma was due to silencing of the EGFR/STAT3 signaling pathway [48]. In this study, we found the expression of p-EGFR and p-STAT3 were attenuated by morin (Appendix A) and the morin/Dox co-treatment in MDA-MB-231 cells (Figure 7), which indicated the co-treatment-induced inhibitions of cell proliferation and death were associated with suppression of the EGFR/STAT3 pathway.

Accumulating evidence indicates that Dox-induced cell cycle arrest at the G2/M phase inhibits cell proliferation and apoptosis in several cancer cell lines [49]. Mechanistically, the induction of G2/M arrest in human breast cancer cells is due to diminished cyclin B1 expression [50]. However, Ling et al. reported that the induction of cyclin B1 by Dox is due to its accelerated synthesis and inhibited degradation, leading to G2/M-phase arrest [51]. Moreover, in human prostate cancer cell lines, cyclin B1 upregulation inhibited cell proliferation and resulted in incomplete cell cycle arrest in the G2/M without entering mitosis [52]. Furthermore, low-dose Dox treatment induced G2/M arrest in Jurkat cells, and this was related to increased cyclin B1 levels, which resulted in a low level of apoptosis and/or mitotic catastrophe [53]. In the present study, Dox treatment primarily induced G2/M arrest (Figure 8A) and increased cyclin B1 levels (Figure 5). Hence, it appears that at different concentrations, Dox could induce G2/M arrest without entering mitosis and inhibiting the degradation of cyclin B1. In contrast, morin/Dox co-treatment decreased cyclin B1 and cyclin D1 expressions and predominantly induced S and G2/M arrest (Figure 8A). Furthermore, we previously reported that morin-induced S and G2/M arrest led to cell death in the MDA-MB-231 cell line [54]. Taken together, these results suggest that the initial induction of S and G2/M arrest by morin/Dox co-treatment may be related to the prominent effect of morin. However, increases in morin/Dox concentration ratio increased the population in the G2/M phase. We surmise that this was associated with morin-induced increased Dox uptake, enhanced Dox cytotoxicity, reduced cell proliferation, and subsequent cell death.

p21 is a major target of p53, leading to cell cycle arrest in response to DNA damage. p21 plays an important role in DNA repair by inhibiting cell cycle progression and facilitating apoptosis [55]. Additionally, transcriptional activation of p21 contributes to the inhibition of colorectal cancer cell growth and the degradation of cyclin D1 [56], which participates in cell cycle progression and can function as a transcriptional co-regulator. Cyclin D1 overexpression is linked with the development and progression of cancer [29], and low cyclin D1 levels inhibit cell cycle progression and, thus, cell proliferation [57]. In this study, the combined morin/Dox treatment upregulated p21 and downregulated cyclin D1 protein levels (Figure 8B), indicating increased DNA damage, cancer cell cycle arrest, and eventually cell death by altering the expressions of p21 and cyclin D1.

This study shows that the pharmacological activity of morin/Dox combinatorial treatment in MDA-MB-231 cells are due to the synergistic enhancement of Dox cytotoxicity, inhibition of the DNA repair system, upregulation of the cellular uptake of Dox, downregulation of FOXM1, and attenuation of the EGFR/STAT3 signaling pathway (Figure 9). These findings suggest that morin has the potential to be used in combination with Doxorubicin for the treatment of TNBC, while also reducing Dox-related side effects. However, further in vivo investigation is required to better understand the therapy and its mechanism.

## 4. Materials and Methods

### 4.1. Materials

Morin, Doxorubicin (Dox), and sulforhodamine B (SRB) were purchased from Sigma-Aldrich (Merck KGaA, Darmstadt, Germany). Dulbecco’s Modified Eagle’s medium (DMEM) and trypsin were obtained from Welgene (Daegu, Republic of Korea), fetal bovine serum (FBS) was from Cytiva (HyClone; Marlborough, MA, USA), and antibiotic/antimycotic solution from Welgene, Inc. (Gyeongsan, Republic of Korea). Polyacrylamide solution (30%), protease inhibitor cocktail, and phosphatase inhibitor cocktail were purchased from GenDEPOT (Katy, TX, USA), and bicinchoninic acid (BCA) protein assay kits and horseradish peroxidase-conjugated goat anti-mouse and -rabbit IgG were from Pierce Biotechnology (Rockford, IL, USA). Trichloroacetic acid (TCA) was purchased from Samchun Pure Chemical Co., Ltd., (Pyeongtaek, Republic of Korea). Sodium dodecyl sulfate (SDS) and N,N,N′,N′-tetramethylethylenediamine were purchased from VWR Life Science AMRESCO Biochemicals (Solon, OH, USA). Primary antibodies for p38 (cat. no. 8690), phospho-p38 (cat. no. 4511), extracellular signal-regulated kinase (ERK; cat. no. 4695), phospho-ERK (cat. no. 4370), c-Jun N-terminal kinase (JNK; cat. no. 9258), phospho-JNK (cat. no. 4668), signal transducer and activator of transcription 3 (STAT3; cat. no. 4904), phospho-STAT3 (cat. no. 9145), epidermal growth factor receptor (EGFR; cat. no. 4267), phospho-EGFR (cat. no. 3777), phospho-H2A histone family member X (p-H2A.X; cat. no. 9718), survivin (cat. no. 2808), FOXM1 (cat. no. 5436), cyclin B1 (cat. no. 4138), cyclin D1 (cat. no. 55506), p21 (cat. no. 2947), poly ADP-ribose polymerase (PARP; cat. no. 9542), caspase-3 (cat. no. 9662), cleaved caspase-3 (cat. no. 9661), caspase-7 (cat. no. 9492), cleaved caspase-7 (cat. no. 9491), B-cell lymphoma-extra-large (Bcl-xL; cat. no. 2764), apoptosis inducing factor (AIF; cat. no. 5318), microtubule-associated protein 1A/1B-light chain 3 (LC3; cat. no. 4108) and, Bcl-2-associated X protein (BAX; cat. no. 2772) were purchased from Cell Signaling Technology (Beverly, MA, USA), and primary antibodies for RAD51 recombinase (RAD51; cat. no. sc-8349), receptor-interacting protein kinase 3 (RIPK3; sc-374639) and β-actin (sc-69879) were obtained from Santa Cruz Biotechnology Inc. (Dallas, TX, USA).

### 4.2. Cell Culture

MDA-MB-231 TNBC cells were obtained from the Korean Cell Line Bank in Seoul, Republic of Korea and cultured in DMEM medium containing 1% antimycotic/antibiotic solution (100 units/mL of penicillin, 100 µg/mL of streptomycin, and 0.25 µg/mL amphotericin B), along with 10% heat-inactivated FBS in a 5% CO_2_ atmosphere at 37 °C. For the experiments, culture media were replaced with media supplemented with 1% antimycotic/antibiotic solution and 2% FBS.

### 4.3. Cytotoxicity Assay

SRB assays were used to evaluate the cytotoxic effects of Dox, morin, and morin/Dox on MDA-MB-231 cells. Briefly, cells (5 × 10^3^) were seeded in a 96-well plate, treated with various concentrations of Dox (0–1 µM), morin (0, 50 and 100 µM), or morin/Dox (morin 0–100 µM and Dox 0.3 µM) after 24 h and incubated for indicated time interval. After that, the cells were fixed using a 10% TCA solution for 1 h, followed by washing with tap water and drying at room temperature (RT). Then, the cells were stained with 0.4% SRB solution and kept at RT for 30 min. After that, they were rinsed with 1% acetic acid, and dried again at RT. Tris-HCl (pH 10.5) buffer (200 µL, 10 mM) was used to dissolve the SRB, and optical densities were measured at 510 nm using a Spectramax M2 spectrophotometer (Molecular Devices, LLC, Sunnyvale, CA, USA).

### 4.4. Doxorubicin Accumulation Assay

Dox cellular uptake was measured by flow cytometry. Cells were seeded in 60 mm discs at 2–4 × 10^5^ cells/disc, allowed to attach for 24 h, co-treated with morin/Dox, detached by trypsinization. The cells were washed twice with PBS (phosphate-buffered saline) and resuspended in PBS containing 2% FBS before subjecting them to flow cytometry (BD FACSCalibur II; BD Biosciences, San Jose, CA, USA).

### 4.5. Combination Index Determinations

Drug synergism studies were conducted using Compusyn Version 1.0 software (Combosyn Inc., Paramus, NJ, USA). Combination indices (CIs, a measure of drug interaction) were determined as described by Chou and Talalay [58]. Where, if CI value less than 1, it indicates synergism, CI equal to 1, it represents an additive effect, and CI greater than demonstrates an antagonistic interaction between morin and Dox. CI values were determined for each dose, and effect levels are presented as affected fractions (Fa). CI values at different Fa levels were plotted using Compusyn software [59].

### 4.6. Cell Cycle Analysis

Cells (1–5 × 10^5^) were treated with Dox (0.3 µM) or co-treated with morin and Dox for 24, 48, or 72 h, incubated at 37 °C, fixed with ice-cold 70% ethanol, suspended in PBS containing 0.1 mg/mL RNase, added 1.0 mL of 40 μg/mL propidium iodide, and incubated at RT for 30 min. Cell cycle analysis by using flow cytometry using BD FACS Calibur II manufactured by BD Biosciences located in San Jose, CA, USA.

### 4.7. Western Blotting

Cells were seeded in 60 mm Petri dishes for 24 h before being co-incubated with a combination of 0.3 µM of Dox and varying concentrations of morin (0 to 100 µM) in DMEM medium containing 2% FBS for 72 h. After incubation, the cells were lysed using radioimmunoprecipitation assay lysis buffer (Biosesang, Seongnam, Republic of Korea) supplemented with protease inhibitor cocktail and phosphatase inhibitor cocktail (GenDEPOT, LLC, Barker, TX, USA), centrifuged at 13,000 rpm for 10 min at 4 °C, and supernatants (whole cell lysates) were stored at −80 °C. Protein concentrations were measured using the BCA method. Same amounts of proteins were separated by 6 to 10% SDS-polyacrylamide gel electrophoresis and transferred to polyvinylidene fluoride membranes (Pall Life Science, Port Washington, NY, USA). Membranes were blocked with either 1% bovine serum albumin (BSA) or 5% non-fat dry milk (Santa Cruz Biotechnology, Inc.) in Tris-buffered saline-Tween (TBS-T, 5 mM Tris-HCl, 150 mM NaCl, and 0.1% Tween-20) and probed with primary antibodies diluted at 1:3000 in 1% BSA or 5% non-fat dry milk in TBS-T overnight at 4 °C. Membranes were then rinsed three times with TBS-T and treated with a HRP-conjugated IgG diluted at 1:5000 in TBS-T for 1 h at RT. Target protein bands were developed using a chemiluminescent substrate and photographed using a Luminescent Image Analyzer LAS-4000 (Fujifilm Corporation, Tokyo, Japan). Densities of target protein bands measured using ImageJ (U.S. National Institutes of Health, Bethesda, MD, USA).

### 4.8. Immunofluorescence Staining

Cells were cultured on glass-bottomed cell culture dishes (cat. no. 30108). Adherent cells were treated with Dox (0.3 µM) with or without morin (50 and 100 µM) for 24 h. the cells were fixed using ice-cold methanol for 4 min, followed by acetone for 2 min. After fixation, the cells were blocked using PBS supplemented with 10% FBS and probed with 1:200 diluted phospho-H2A.X antibody in PBS overnight at 4 °C. The cells were then treated with 1:200 diluted Alexa 488 (Green)-conjugated goat anti-rabbit IgG in PBS for 2 h at RT in the dark, stained with 4′,6-diamidino-2-phenylindole, and photographed under a fluorescence microscope (Carl Zeiss, Jena, Germany).

### 4.9. Analysis of Apoptotic and Necrotic Cells

To quantify apoptotic and necrotic cells, Annexin V/7-amino-actinomycin D (7-AAD) (BioLegend) dual staining was performed. Cells were co-treated with Dox with or without morin, washed twice with cold PBS, and resuspended in Annexin V binding buffer at 0.25 × 10^7^ cells/mL for 15 min at RT, as per the manufacturer’s instructions. Samples were analyzed by flow cytometry.

### 4.10. Statistical Analysis

The statistical analysis was carried out using SPSS Ver. 20.0 (SPSS Inc., Chicago, IL, USA). To determine the significance of differences between groups, one-way analysis of variance (ANOVA) and Tukey’s post hoc test were performed. The experiments were performed repeated three times independently, and the results are presented as the means ± standard deviations (SDs). A *p*-value of less than 0.05 was considered statistically significant.

## Figures and Tables

**Figure 1 pharmaceuticals-16-00672-f001:**
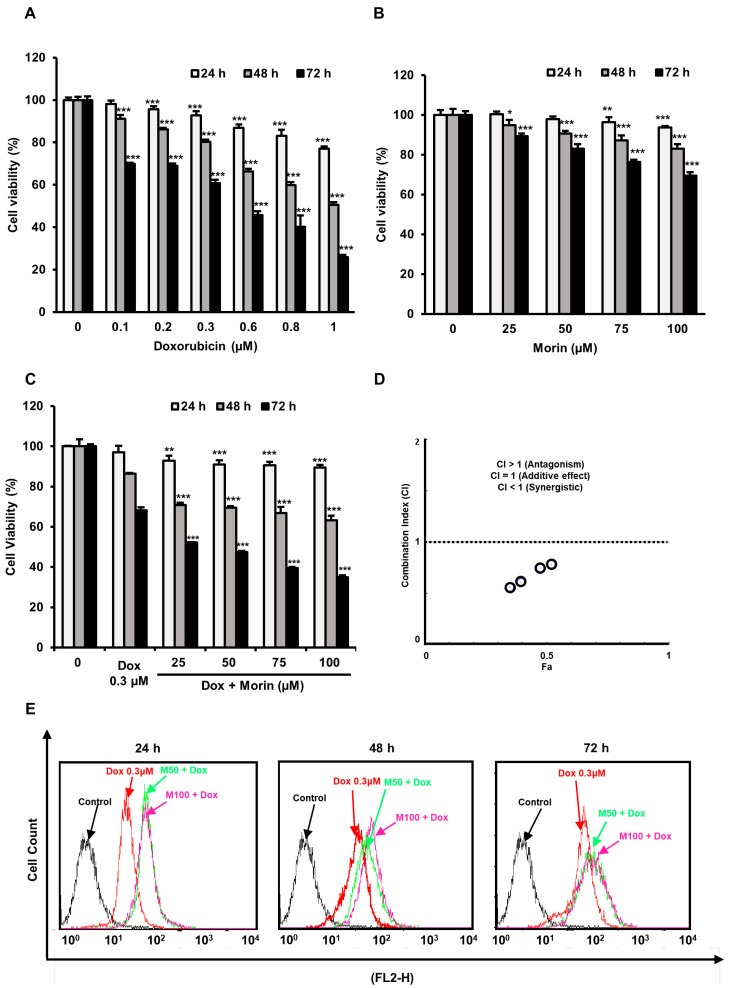
Cytotoxic activity of Doxorubicin was enhanced by morin/Dox co-treatment, and this was associated with Dox uptake by MDA-MB-231 cells. (**A**–**C**) SRB assay was used to measure cell viability. (**A**,**B**) *, ** and *** indicate *p* < 0.01, *p* < 0.001 and *p* < 0.0001 (control vs. treatment), respectively. (**C**) ** and *** indicate *p* < 0.001 and *p* < 0.0001 (Dox vs. Dox + morin), respectively. (**D**) A combination index (CI) versus fractional affected plot for morin/Dox co-treatment (obtained using Compusyn software). Circles that represent CI values. Morin/Dox co-treatment exhibited a synergistic effect with respect to Dox. (**E**) Morin/Dox co-treatment accelerated Dox uptake by MDA-MB-231 cells. Dox: Doxorubicin.

**Figure 2 pharmaceuticals-16-00672-f002:**
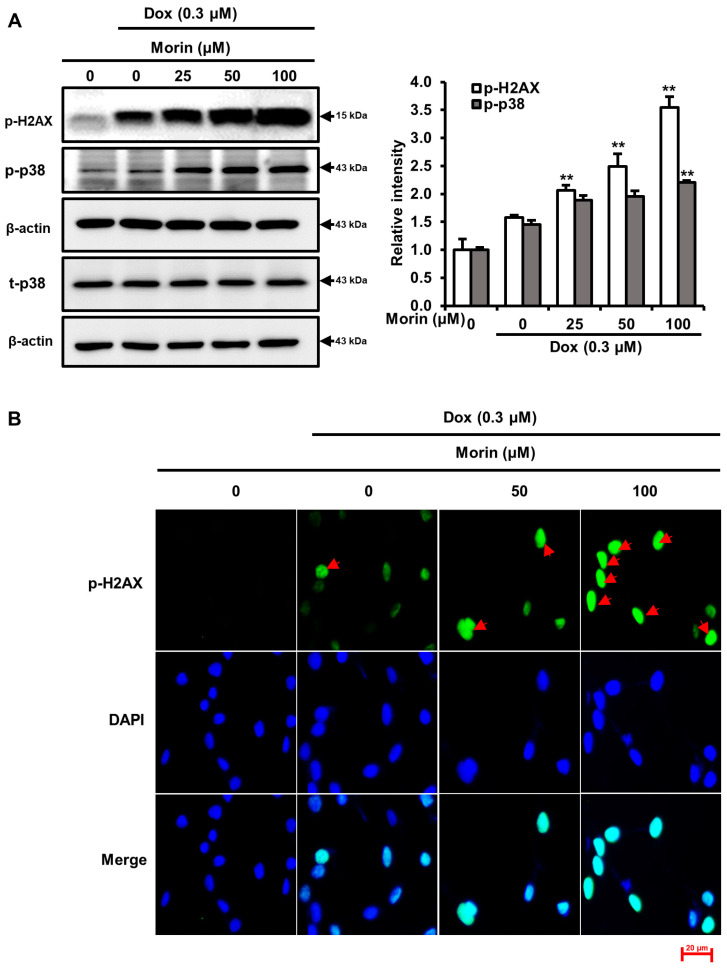
Effects of morin/Dox co-treatment on phosphorylated H2A.X and MAPK proteins expression, and the formation of nuclear H2A.X foci. (**A**,**C**) Relative intensities of bands were evaluated. ** indicates *p* < 0.001 (control vs. treatment). (**B**) Formation of nuclear foci of H2A.X by Dox treatment with or without morin. Cells were visualized under a fluorescence microscope. Red arrow indicated the number of foci formed. Dox: Doxorubicin.

**Figure 3 pharmaceuticals-16-00672-f003:**
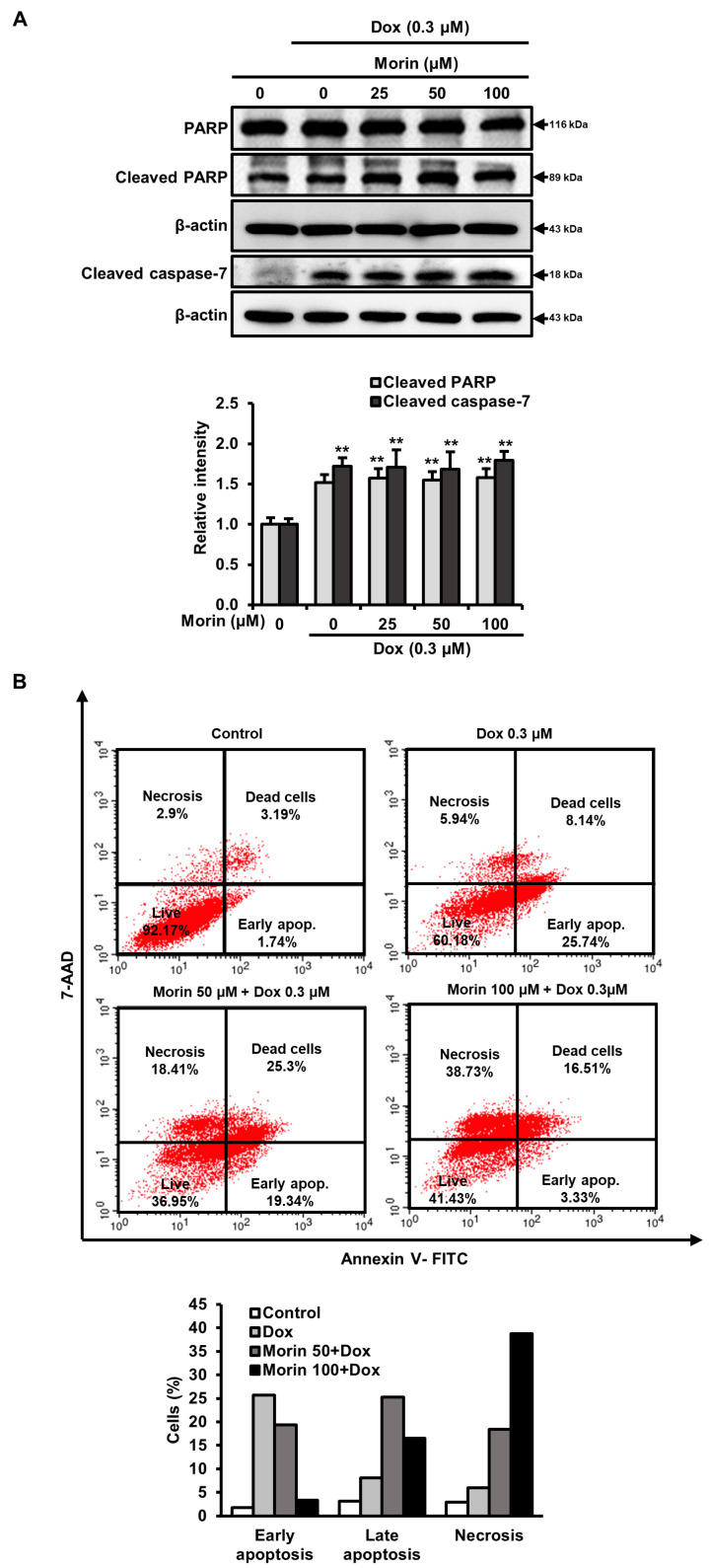
Morin/Dox co-treatment induced cell death. (**A**) The intensity of the band was measured. ** indicates *p* < 0.001 (control vs. treatment). (**B**) The necrotic or apoptotic MDA-MB-231 cell population was assessed by Annexin V/7-AAD dual staining using flow cytometry and represented in bar diagram.

**Figure 4 pharmaceuticals-16-00672-f004:**
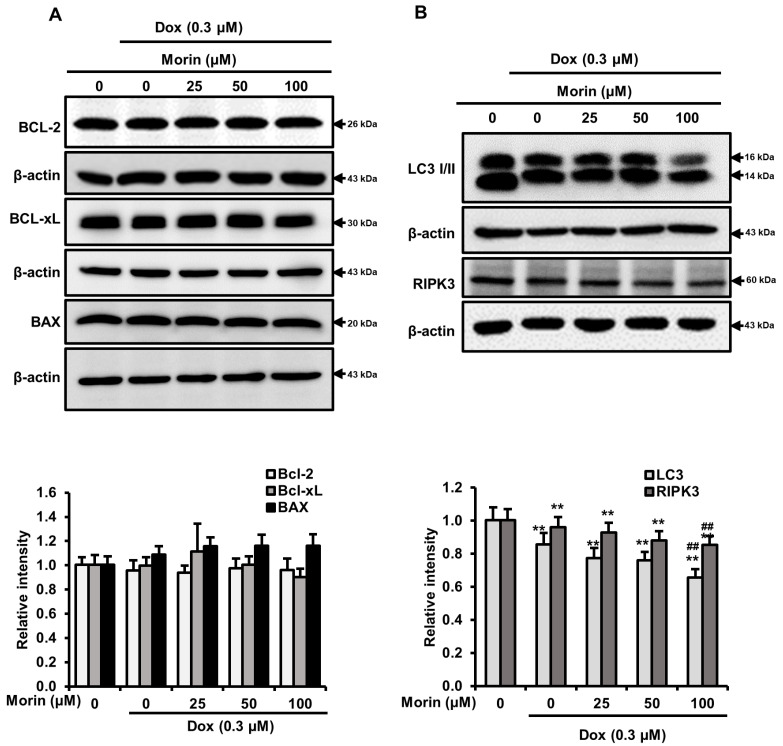
Effects of morin/Dox co-treatment on Bcl-2 protein family proteins, LC3-I/II, and RIPK3. (**A**,**B**) The density of the bands was quantified. ** indicates *p* < 0.001 (control vs. treatment); ^##^ indicates *p* < 0.001 (Dox vs. Dox + morin). Dox: Doxorubicin.

**Figure 5 pharmaceuticals-16-00672-f005:**
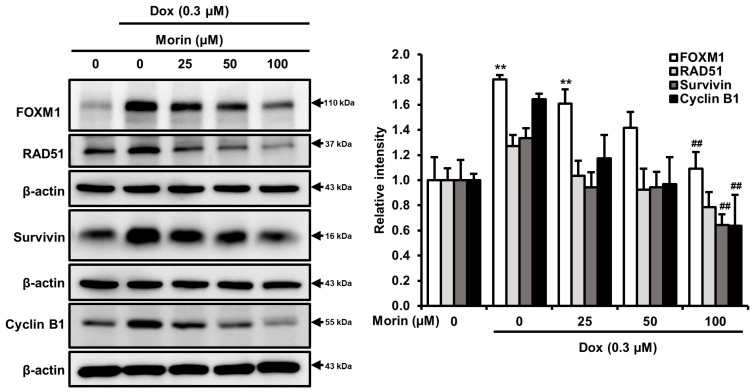
Morin/Dox co-treatment attenuated Dox-induced FOXM1, RAD51, survivin, and cyclin B1 protein expressions: band density was measured. ** indicates *p* < 0.001 (control vs. treatment); ^##^ indicates *p* < 0.001 (Dox vs. Dox + morin). Dox: Doxorubicin.

**Figure 6 pharmaceuticals-16-00672-f006:**
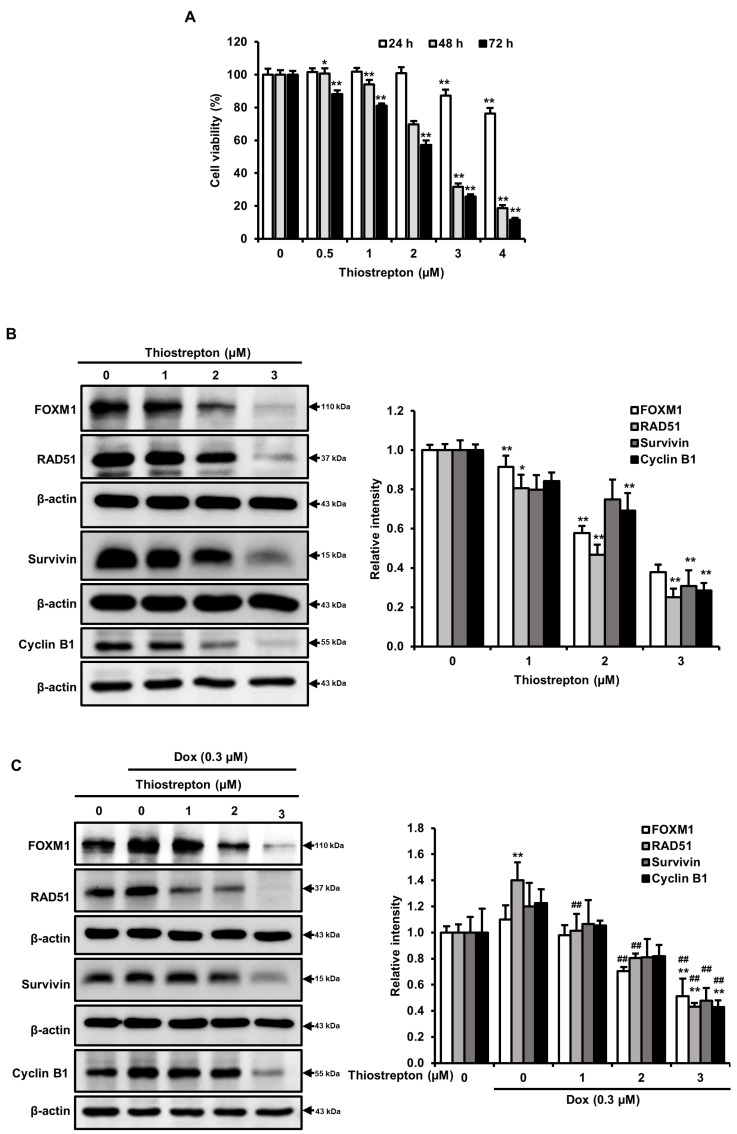
Morin/Dox co-treatment-induced cell death was associated with the FOXM1 pathway: (**A**) Cell survival was assessed by SRB assay. (**B**,**C**) Protein levels were quantitatively evaluated. * and ** indicate *p* < 0.01 and *p* < 0.001 (control vs. thiostrepton), respectively. ^##^ indicates *p* ≤ 0.001 (Dox vs. Dox + thiostrepton). Dox: Doxorubicin.

**Figure 7 pharmaceuticals-16-00672-f007:**
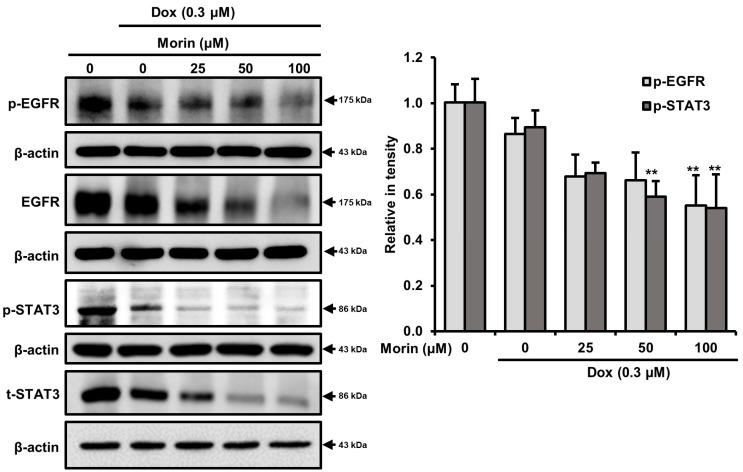
Morin/Dox co-treatment caused cell death by repressing the EGFR/STAT3 pathway. EGFR and STAT3 protein levels quantitatively analyzed. ** *p* < 0.001 (control vs. treatment). Dox: Doxorubicin.

**Figure 8 pharmaceuticals-16-00672-f008:**
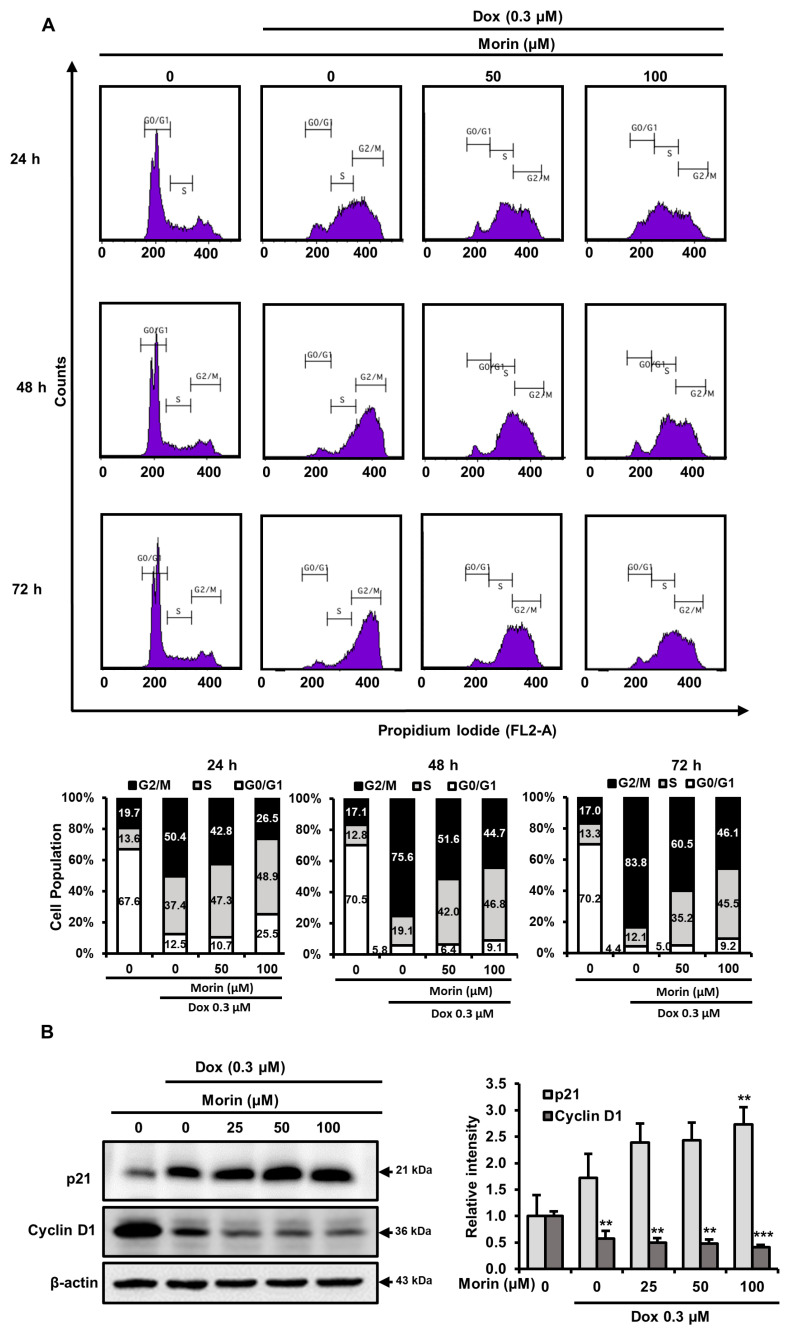
Effects of morin/Dox co-treatment on cell cycle proteins and cycle progression. (**A**) Percentage values in the histogram cell show accumulations in the G0/G1, S, and G2/M phases. Experiments were performed thrice, and results are presented as the means. (**B**) The band densities of p21 and cyclin D1 were measured. ** *p* < 0.001, and *** *p* < 0.0001 (control vs. treatment). Dox: Doxorubicin.

**Figure 9 pharmaceuticals-16-00672-f009:**
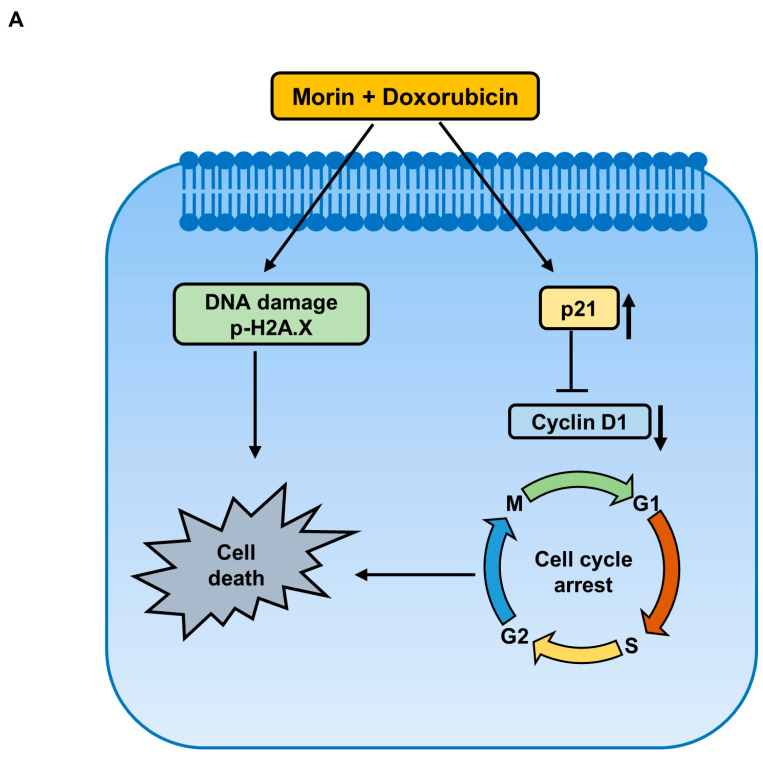
Proposed signaling pathways elucidating the anti-cancer effect of a combined treatment of morin and Doxorubicin in MDA-MB-231 cells. (**A**) Co-treatment with morin and Doxorubicin induces DNA damage and alters cell cycle regulators, leading to cell death. (**B**) The combination treatment inhibits cell survival via two pathways: firstly, by inhibiting FOXM1, which decreases the protein levels of homologous recombination DNA repair proteins, RAD51 and survivin, and the cell cycle protein, cyclin B1; and secondly, by suppressing the EGFR/STAT3 pathway.

## Data Availability

Data is contained within the article and Appendix A.

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
