# Peer review of "Morin Sensitizes MDA-MB-231 Triple-Negative Breast Cancer Cells to Doxorubicin Cytotoxicity by Suppressing FOXM1 and Attenuating EGFR/STAT3 Signaling Pathways"

_pharmaceuticals, 2023, doi:10.3390/ph16050672_

Round 1

Reviewer 1 Report

Maharajan et al studied the combinational effect of Morin and Doxorubicin on MDA-MB-231 cells. They showed that Dox/Morin induce cell death via necrosis and suppresses FOXM1 and EGFR/STAT3 pathways in MDA-MB-231 cells. The authors efforts are commendable, however there are some significant issues that need to be addressed for the manuscript to be considered for publication.

Major Comments:

1.     In Figure2, include a sample with only Morin treatment to confirm that the effect seen is indeed due to synergistic effect by DOX/Morin and not just by Morin alone, like the authors have shown in Figure 1. Its obvious that the inhibition of cell viability is due to DOX/Morin.

2.     In Figure4A, the authors claim that the co-treatment of DOX/Morin did not affect Bcl-xL, Bcl-2, and BAX. But the blots show that these proteins are reduced with the co-treatment. Therefore, need a better representative image. Include a sample treated with just Morin alone.

3.     As the authors are referring strongly that DOX/Morin promoted cell death via necrosis, it would be better if they try to detect other necrotic markers like HMGB1, LDH and HSP70 and show that their expression goes down with the combinational treatment to make their statement stronger. Include a sample treated with just Morin alone.

4.     The authors claim that the cell viability went down when cells were treated with DOX/Morin and thiostrepton in Figure6B. But the figure depicted tells otherwise, especially at 48h.

5.     Figure6D doesn’t show the combination effects of DOX and thiostrepton. It looks like the effect is happening only with thiostrepton as seen in Figure6C. The authors need to run the samples from individual treatment of thiostrepton and combinational treatment of thiostrepton with Dox on the same membrane to confirm their claim. Also have to include DOX/Morin.

6.     Include a sample with Morin alone in Figure7 and Figure8.

7.     All the western blots must have samples treated with Morin alone too because the authors want to show the synergistic effect of DOX/Morin. Therefore, it is must to have the samples from all individual and combinational treatments run on the same membrane.

Minor Comments:

1.       # missing for RAD51 graph in Fig5.

2.       Mention the treatment time for all the westerns.

3.       The superscript of reference numbering is not consistent. Example, line 31 and 33.

4.       Adjust the labelling in Figure6D and 8A.

Author Response

We revised the manuscript using track change.

We replied to comments using blue-colored text; added supplementary data Fig S1; revised figures 4A, 6, 8.

Reviewer 2 Report

In the article 'Morin sensitizes MDA-MB-231 triple-negative breast cancer cells to doxorubicin cytotoxicity by suppressing FOXM1 and attenuating EGFR/STAT3 signaling pathways'  Maharjan S et al., the authors investigate the combinatorial effect of Doxycycline and Morin, a flavonoid that is originally isolated from the Moraceae family, on tripple negative breast cancer cells. While the combination treatment synergistically reduce the viability, cell cycle progression, the main drawback of the study is the lack of mechanistic insight of the action of this phytochemical Morin. Or in other words the authors fails to show what is the specific target of Morin in this study. The combination treatment seem to be affecting a lot of signaling pathways including EGFR-STAT3, FOXM1, p21 etc. One of main reason behind drug toxicities is lack of specificity. Also, the authors claim that the combination might reduce Dox-related side effects. This needs to be supported by evidence.

I suggest the authors to answer the following questions in order to support their claims -

1. How does the combination vs single dox treatment affect the proliferation of normal non-tumorigenic cells. for example MCF10?

2. Upon loss of function of FoxM1 or after treatment with Thiostrepton, are the levels of pSTAT3, pEGFR, p21 affected? Are these pathways related?

3. Can you rescue the effect of the combination on cell survival by overexpressing RAD51 and Survivin?

Author Response

We replied to comments with blue-colored text.

Reviewer 3 Report

In this manuscript, authors showed that improved therapeutic efficacy of Morin in presence of chemotherapeutic drug, Doxorubicin against TNBC cell line, MDA-MB-231. Co-treatment of morin/Dox enhanced DNA damage and P38 activation, cytotoxic effect and necrotic cell death. Moreover, combination treatment of morin and Dox inhibited EGFR/STAT3 pathway in MDA-MB-231 cells. Further, FOXM1 pathway and cell cycle progression was attenuated with combination treatment of morin and Dox that led to increased cell death. Overall this study is very well planned and accomplished by utilizing in vitro experimental approaches.

Major Comments:

1. This study clearly lacks an in vivo mouse model studies of TNBC to further prove the enhanced anti-tumor efficacy of morin and Dox.

2. The major drawback of this study is utilizing one cell line (MDA-MB-231). It would be important to utilize 2 or more cancer cell lines to show the anti-tumor activity of drugs.

3. Although, morin was widely studied in different tissue specific cancers, authors would show the level of cytotoxic effect of morin and Dox on non-tumorigenic cells.

Minor comments:

1. Correct grammatical mistakes and typo errors.

Author Response

We revised the manuscript using track change. We replied to comments with blue-colored text.

Reviewer 4 Report

The article "Morin sensitizes MDA-MB-231 triple-negative breast cancer cells to doxorubicin cytotoxicity by suppressing FOXM1 and attenuating EGFR/STAT3 signaling pathways" is an interesting scientific paper. Despite the fact that the methodology is based on an in vitro system, after conducting in vivo tests, the obtained results can be used in clinical diagnostics.                                                                                                    The layout of the article is typical for this kind of scientific studies.             The readability of all figures and the included graphic abstract (figure 9) deserve to be distinguished.                                                                        Due to the significant importance of the research results in the scientific and perhaps clinical aspect, I propose to publish the article " Morin sensitizes MDA-MB-231 triple-negative breast cancer cells to doxorubicin cytotoxicity by suppressing FOXM1 and attenuating EGFR/STAT3 signaling pathways' in PHARMACEUTICALS in the present form.

                                                                        .

Author Response

We revised the manuscript using track change. We replied to comments with blue-colored text; revised Figure 9.

Round 2

Reviewer 1 Report

I am satisfied with the authors revisions.

Author Response

Thank you for the comment.

Reviewer 2 Report

In this study authors could show Morin acts in synergy with Doxorubicin by downregulating FOXM1 and STAT3 pathways. The could address whether Morin would be protective towards normal cells and hints that the combination could be beneficial over using dox as monotherapy. They also addressed whether Morin independently affected FOXM1 and Stat3 pathways. In future, in vivo experiments can potentially consolidate the authors claim. 

Author Response

Thank you for the comment. 

Reviewer 3 Report

I recommend this manuscript for publication in present form.

Author Response

Thank you for the comment.